# Control-ITRA: Controlling the Behavior of a Driving Model

## Abstract

Simulating realistic driving behavior is crucial for developing and testing autonomous systems in complex traffic environments. Equally important is the ability to control the behavior of simulated agents to tailor scenarios to specific research needs and safety considerations. This paper extends the general-purpose multi-agent driving behavior model ITRA (Ścibior et al., 2021), by introducing a method called Control-ITRA to influence agent behavior through waypoint assignment and target speed modulation. By conditioning agents on these two aspects, we provide a mechanism for them to adhere to specific trajectories and indirectly adjust their aggressiveness. We compare different approaches for integrating these conditions during training and demonstrate that our method can generate controllable, infraction-free trajectories while preserving realism in both seen and unseen locations.

## 1 Introduction

The simulation of realistic driving behavior is a cornerstone in the development and validation of autonomous driving systems. As autonomous vehicles (AVs) increasingly integrate into real-world traffic, the necessity for robust, reliable, and diverse simulation environments becomes paramount. These environments enable the testing of AVs in complex, high-stakes scenarios that would be difficult or dangerous to replicate in real-world conditions. Moreover, the ability to simulate realistic multi-agent interactions is critical for ensuring that AVs can navigate and respond appropriately to the unpredictable behavior of human drivers and other road users.

One of the key challenges in multi-agent driving simulations is the balance between realism and control. State-of-the-art models (Ścibior et al., 2021; Suo et al., 2021; Nayakanti et al., 2023; Gulino et al., 2023; Seff et al., 2023; Wu et al., 2024) aim to replicate the nuances of human driving behavior but often lack the flexibility to adapt to specific research needs or safety protocols. The ability to control the behavior of simulated agents is essential for tailoring scenarios to investigate particular driving conditions, test edge cases, or enforce safety standards. However, introducing control mechanisms without sacrificing realism remains a significant challenge in the field.

Conceptually, human driving behavior encompasses numerous unobserved variables, ranging from high-level goals such as "going to the grocery store across the roundabout," to intermediate behavioral traits like aggressiveness, down to low-level controls such as setting acceleration and steering values. An ideal driving simulator would allow conditioning on any of these variables, enabling targeted scenario design. However, achieving such comprehensive control is challenging due to the difficulty of precisely defining different behaviors or measuring the degree to which conditions are met.

In this work, we introduce Control-ITRA, a model that enables the control of agent behavior through two primary methods: by specifying waypoints for the agent to follow and by setting a target speed for it to reach. Waypoints provide a natural mechanism for guiding agents along a desired path, while target speeds offer a way to influence the agent's aggressiveness indirectly. Specifically, we build upon the ITRA framework (Ścibior et al., 2021), a state-of-the-art model that leverages rasterized overhead birds-eye view representations to perceive its environment. We selected ITRA as our foundation, as birdviews offer an intuitive means of spatially placing waypoints. Additionally, we developed a mechanism to assign target speeds per agent, supporting both conditional and unconditional control execution.

We further explore two strategies for selecting conditions during training, demonstrating that our approach, Control-ITRA, enables the model to meet specified conditions while preserving the realism of the driving behavior. Finally, we evaluate our conditional model on unseen, out-of-domain locations using TorchDriveEnv (Lavington et al., 2024), a reinforcement learning environment with simulated traffic, and show that our method outperforms traditional reinforcement learning baselines in the benchmark validation scenarios from TorchDriveEnv.

## 2 Related Work

**Trajectory Prediction:** Numerous advanced autonomous vehicle simulators have been proposed in recent years (Dosovitskiy et al., 2017; Santara et al., 2021; Zhao et al., 2024), reflecting the community's growing recognition of simulation as an essential element for achieving Level 5 autonomous driving (On-Road Automated Driving (ORAD) Committee, 2021). In this paper, we focus on trajectory prediction models that can simulate realistic traffic behavior. The primary task of trajectory prediction models is to predict future trajectories based on observed environmental behavior. Broadly, trajectory models can be classified into physics-based and learning-based models. Physics-based methods leverage physical models to generate trajectories with relatively low computational resources, often using kinematic and dynamic models (Lin & Ulsoy, 1995; Lytrivis et al., 2008; Brännström et al., 2010) combined with inference techniques like Kalman Filters (KF) (Ammoun & Nashashibi, 2009; Jin et al., 2015; Lefkopoulos et al., 2021) and Monte Carlo methods (Althoff & Mergel, 2011; Okamoto et al., 2017; Wang et al., 2019). These traditional methods are generally suitable only for simple prediction tasks and environments.

Recently, deep learning-based methods have gained popularity due to their ability to model complex physical, road-related, and agent-interactive factors, making them adaptable to more realistic environments. Predicting future states is inherently probabilistic, and methods like those in Cui et al. (2019); Chai et al. (2020) forecast multiple possible trajectories for each agent. Djuric et al. (2020) employs rasterized ego-centric and ego-rotated birdview representations to depict an agent's current and past states, using a CNN to predict future trajectories. Similarly, ITRA (Ścibior et al., 2021) uses ego-centric birdview representations to perceive the environment, modeling each agent as a variational recurrent network (Chung et al., 2015). Tang & Salakhutdinov (2019) applies a discrete latent model with a fixed number of future trajectories per agent, utilizing a different representation with separate modules for map encoding and individual RNN networks for encoding agent states. Casas et al. (2020) leverages spatially-aware graph neural networks to model agent interactions in the latent space. Transformer-based approaches (Liu et al., 2021; Huang et al., 2022; Seff et al., 2023; Niedoba et al., 2023; Wu et al., 2024) have also been widely adopted to encode interactions between agent states.

**Goal-conditioned Models:** In the literature, conditioning on waypoints is typically framed as a goal-conditioning task, often addressed through inverse planning. Here, trajectory prediction is divided into first predicting candidate waypoints and then generating trajectories based on these waypoints. PRECOG (Rhinehart et al., 2019) introduces a probabilistic forecasting model conditioned on agent positions. PECNet (Mangalam et al., 2020) generates endpoints for pedestrian trajectory prediction in a two-step process, where the proposed endpoints guide pedestrian trajectory sequences. Graph-TERN (Bae & Jeon, 2023) divides pedestrian future paths into three sections, inferring a goal point for each section using mixture density networks. MUSE-VAE (Lee et al., 2022) uses a conditional VAE model to generate short-term and long-term goal heatmaps, from which the agent trajectory is then conditioned. DenseTNT (Gu et al., 2021) predicts a dense goal probability distribution over the road ahead and uses a goal set prediction model to determine the final trajectory goals. Y-net (Mangalam et al., 2021) generates goal position heatmaps using a convolution-based approach, sampling final endpoints from the resulting goal distribution. In Goal-LBP (Yao et al., 2024), goal endpoints are generated based on both static context maps and dynamic local behavior information. S-CVAE (Zhang et al., 2024) reformulates point prediction as a region-generation task, constructing an incremental greedy region to enlarge the coverage of candidate waypoints allowing to model the multimodality of behavioral intentions. CTG (Zhong et al., 2023b) employs a scene diffusion model that predicts the whole multi-agent trajectory sequence all at once allowing for waypoint conditioning using diffusion guidance. In a similar fashion, Safe-Sim (Chang et al., 2024) uses diffusion guidance to direct agents

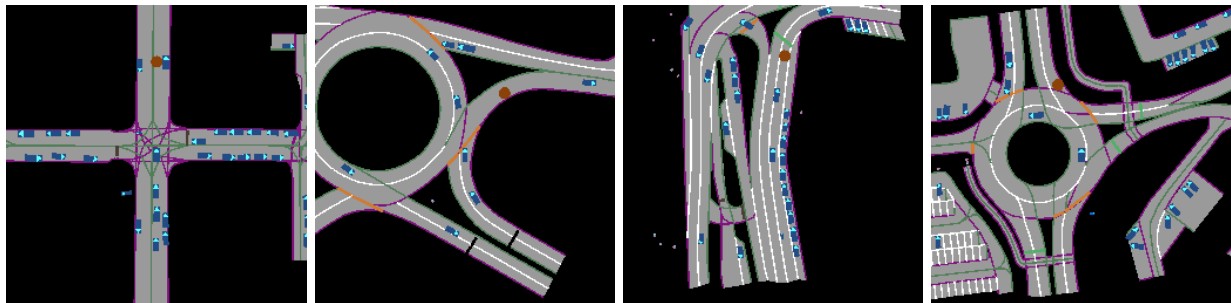

Figure 1: Example ego-centric and ego-rotated birdview representations from various locations in the training set. Waypoints are shown as brown circles.

on predefined agent route paths on a lane graph from their starting point to their destination. SceneDiffuser (Jiang et al., 2024) also uses a diffusion model that allows for controllability by pre-filling at the start of the diffusion process the conditioned agent trajectories with positions to be reached. These positions can be manually specified or generated using a language model and a predefined structured data format. MixSim (Suo et al., 2023) is a multi-agent driving policy that explicitly requires conditioning all agents in the scene to goals expressed as directed paths of lane segments composed of sequences of roadgraph nodes. This differs from our definition of goals expressed as waypoints which allows placing waypoints anywhere on the map independently of the roadgraph. The authors propose multiple ways to sample conditions at test time to simulate desired scenarios. We argue that any method that extracts goal conditions directly from the ground truth data can benefit from our proposed Control-ITRA sampling scheme for extracting training conditions.

CtRL-Sim (Rowe et al., 2024) follows a different learning paradigm, using offline reinforcement learning to train an agent and a reward function that includes a specified goal position satisfaction. Other forms of controllability include CTG++ (Zhong et al., 2023a) which describes a method for generating diffusion guidance objectives using scene goals expressed in natural language. Finally, Vista (Gao et al., 2024) employs a different approach, learning a driving world model using video diffusion from the driver's first-person view, where waypoint conditioning is achieved by selecting a 2D coordinate projected from the ego vehicle's short-term destination onto the initial frame.

Unlike previous work, our method does not focus on generating goal waypoints at inference time. Instead, we concentrate on developing a driving behavior model that can realistically follow either densely or sparsely placed waypoints by effectively amortizing (Lioutas et al., 2022) the distribution of waypoint-conditional driving behavior extracted from human traffic data. In addition, we introduce a second type of controllability in the form of target speeds, which can implicitly allow us to vary driving aggressiveness. SCBG (Chang et al., 2023) describes an alternative formulation for driving aggressiveness by attempting to quantify courtesy between two driving agents and condition on these values.

## 3 Method

In this section, we first introduce the driving behavior model that serves as our foundation model for controlling its behavior. We then explain how to introduce a conditional variable and suggest Control-ITRA a training scheme for learning such conditional driving behavior models. Finally, we propose two types of conditioning for controlling driving agents.

### 3.1 Background: ITRA

The main contribution of this paper is to enable the control of a driving behavior model by conditioning its output. Doing so will allow the extraction of interesting interactive behaviors that can be used for testing and further improving driving models. Numerous generative models have been proposed in the literature (Ścibior et al., 2021; Suo et al., 2021; Nayakanti et al., 2023; Gulino et al., 2023; Seff et al., 2023; Niedoba et al., 2023; Wu et al., 2024). We select ITRA (Ścibior et al., 2021) as our base model, a driving behavior

model trained on real-world traffic data that provides a convenient representation of the observed world state.

In ITRA, the environment is represented as a rasterized birdview image encoding interactions between the ego agent, other agents, and the surrounding environment. These ego-centric, ego-rotated birdview images are denoted as $b_t^i \in \mathbb{R}^{H \times W \times 3}$ for each agent $i$ and timestep $t$, and they are generated using a rendering function $b_t^i = \texttt{render}(i, s_t^{1:N}, V)$ where $V$ is a triangle mesh representing the drivable area. A trajectory segment is represented as a sequence of states $s_{1:T} = \{s_1^{1:N}, \ldots, s_T^{1:N}\}$, where $T$ is the number of timesteps and $N$ is the number of agents in the segment. Each state is a tuple $s_t^i = (x_t^i, y_t^i, \psi_t^i, v_t^i) \in \mathbb{R}^4$, where $x_t^i$ and $y_t^i$ denote the coordinates of the agent's geometric center, $\psi_t^i$ represents its orientation, and $v_t^i$ its current speed. Each agent is represented as a rotated bounding box with length $l^i$ and width $w^i$, which are assumed to be provided.

ITRA is structured as a multi-agent variational recurrent neural network (VRNN) (Chung et al., 2015) where each agent samples its own latent variables $z_t^i$. The generative model is followed by a standard bicycle kinematic model (Rajamani, 2012; Ścibior et al., 2021), which transforms each agent's actions $a_t^i = (\alpha_t^i, \beta_t^i)$ into the next state $s_{t+1}^i$, where $\alpha_t^i$ represents the acceleration and $\beta_t^i$ the steering angle. The joint distribution of ITRA is given by

$$p_\theta(s_{1:T}) = p_0(s_1^{1:N})p_0(h_0^{1:N}) \int \int \prod_{t=1}^{T} \prod_{i=1}^{N} p(z_t^i)p(b_t^i|i, s_t^{1:N}, V)p_\theta(a_t^i|b_t^i, z_t^i, h_{t-1}^i) \tag{1}$$
$$p_\theta(h_t^i|h_{t-1}^i, a_t^i, b_t^i, z_t^i)p(s_{t+1}^i|s_t^i, a_t^i)dz_{1:T}^{1:N}da_{1:T}^{1:N},$$

where $p_0(s_1^{1:N})$ is a given distribution of initial states, $p_0(h_0^{1:N})$ is the distribution of initial recurrent states and

$$p(z_t^i) = \mathcal{N}(z_t^i; 0, \mathbf{I}), \tag{2}$$
$$p(b_t^i|i, s_t^{1:N}, V) = \delta_{\texttt{render}(i, s_t^{1:N}, V)}(b_t^i), \tag{3}$$
$$p_\theta(a_t^i|b_t^i, z_t^i, h_t^i) = \mathcal{N}(a_t^i; \mu_\theta(b_t^i, z_t^i, h_{t-1}^i), \mathbf{I}), \tag{4}$$
$$p_\theta(h_t^i|h_{t-1}^i, a_t^i, b_t^i, z_t^i) = \delta_{\texttt{RNN}_\theta(h_{t-1}^i, a_t^i, b_t^i, z_t^i)}(h_t^i), \tag{5}$$
$$p(s_{t+1}^i|s_t^i, a_t^i) = \delta_{kin(s_t^i, a_t^i)}(s_{t+1}^i). \tag{6}$$

The model is optimized using the standard evidence lower bound objective (ELBO). This process minimizes the negative ELBO, defined as

$$\mathcal{L}_{\text{ELBO}} = \mathop{\mathbb{E}}_{s_{1:T} \sim p_D(s_{1:T})} \Big[ \sum_{t=1}^{T-1} \sum_{i=1}^{N} \Big( \mathop{\mathbb{E}}_{q_\phi(z_t^i|a_t^i, b_t^i, h_{t-1}^i)} \big[\log p_\theta(s_{t+1}^i|b_t^i, z_t^i, h_{t-1}^i)\big] - D_{\text{KL}} \big[q_\phi(z_t^i|a_t^i, b_t^i, h_{t-1}^i)||p(z_t^i)\big] \Big) \Big] \tag{7}$$

$$\leq \mathop{\mathbb{E}}_{s_{1:T} \sim p_D(s_{1:T})} \Big[ \log p_\theta(s_{1:T}) \Big]$$

where the recurrent states $h_{1:T}$ are generated using the RNN network from Equation (5), $q_\phi$ is a separate inference network approximating the proposal distribution defined as

$$q_\phi(z_t^i|a_t^i, b_t^i, h_{t-1}^i) = \mathcal{N}(z_t^i; \{\mu_\phi, \sigma_\phi\}(a_t^i, b_t^i, h_{t-1}^i)), \tag{8}$$

and trained jointly with the model $p_\theta$.

## 3.2 Training with Conditions

We aim to obtain a driving behavior model that can drive vehicles realistically while optionally following agent-specific conditions. In this section, we introduce the principal way of training such conditional models. Specifically, we extend the main training procedure of ITRA (Ścibior et al., 2021) to utilize the additional

---

**Algorithm 1** Conditional ITRA Training Step

---
    **Input:** Ground truth segment $s_{1:T}^{1:N}$
               Ego-agent index $i$
               Behavior model $p_\theta$
               Ordered list of conditions $C$
               Conditioning probability $p_C$
    **Output:** Total loss $\mathcal{L}_{\text{ELBO}}$
1:  *use_condition* $\leftarrow$ randomly enable conditioning with probability $p_C$
2:  $\mathcal{L}_{\text{ELBO}} \leftarrow 0$
3:  $k \leftarrow 1$
4:  **for** $t \in 2 \ldots T$ **do**
5:     **if** *use_condition* and $k \leq \text{len}(C)$ **then**
6:        $\hat{s}_t^i \sim p_\theta(s_t^i | s_{1:t-1}^{1:N}, C_k)$ using proposal distribution $q_\phi$
7:        **if** $C_k$ is reached **then**
8:           $k \leftarrow k + 1$
9:     **else**
10:       $\hat{s}_t^i \sim p_\theta(s_t^i | s_{1:t-1}^{1:N}, \varnothing)$ using proposal distribution $q_\phi$
11:    $\mathcal{L}_{\text{ELBO}}^t \leftarrow$ compute using $s_t^i$ and $\hat{s}_t^i$
12:    $\mathcal{L}_{\text{ELBO}} \leftarrow \mathcal{L}_{\text{ELBO}} + \mathcal{L}_{\text{ELBO}}^t$
13: **return** $\mathcal{L}_{\text{ELBO}}$

---

conditions. We refer to these new conditional models $p_\theta(s_t^i | s_{1:t-1}^{1:N}, C_k)$ as Control-ITRA where $C_k$ is the condition given at timestep $t$ for agent $i$. We assume that by design the conditional behavior model allows for an optional passing of an agent condition (i.e. $C_k = \varnothing$) which in this case the model should default to unconditional prior behavior (i.e. $p_\theta(s_t^i | s_{1:t-1}^{1:N}, \varnothing) := p_\theta(s_t^i | s_{1:t-1}^{1:N})$) for the predicted $i$ agent at timestep $t$. Algorithm 1 describes the process of executing a single step for training a conditional model. Given a ground truth sequence of states $s_{1:T}^{1:N}$ for $N$ agents and an ordered list of conditions for the ego-agent, the conditioning for the current training step is enabled with a probability $p_C$. The use of the conditioning probability $p_C$ allows for training both conditionally and unconditionally using a single behavioral model. During each timestep $t$ within the training segment length $T$, the model predicts the ego state $\hat{s}_t^i$ conditioned on the previous states $s_{1:t-1}^{1:N}$ and the current condition $C_k$ if conditioning is enabled. The transition to the next condition occurs if the current condition is reached according to the condition type. If conditioning is not enabled, the model predicts the state without any conditional information. The algorithm iteratively computes the evidence lower bound loss $\mathcal{L}_{\text{ELBO}}^t$ for each timestep by comparing the predicted state $\hat{s}_t^i$ to the ground truth $s_t^i$. The total loss $\mathcal{L}_{\text{ELBO}}$ accumulates over all timesteps.

### 3.3 Waypoint Conditioning

An intuitive way of controlling the behavior of the simulated agents is to set waypoints for them to follow. Specifically, we formally define waypoints $w_{1:K_i}^i$ for each agent $i$ as an ordered collection of $K_i$ tuples of target coordinates where $w_k^i = (x_k^i, y_k^i)$. Additionally, a waypoint is considered *reached* from an agent $i$ at a timestep $t$ when

$$\sqrt{(x_t^i - x_k^i)^2 + (y_t^i - y_k^i)^2} \leq R, \tag{9}$$

where $R$ is a hyperparameter and corresponds to the radius from the center of the waypoint. In our definition of the waypoint following task, the agent must reach each waypoint sequentially in the specified order. Once a waypoint is deemed reached, the next waypoint in the sequence is shown. Each agent is presented with only one waypoint at any time from the waypoints list $w_{1:K_i}^i$.

The agents are not constrained to reach waypoints as quickly as possible or within a specific timeframe. Instead, they are free to take any actions necessary to reach the target point safely and realistically. Waypoints that cannot be reached safely should be ignored. Finally, waypoints are an optional condition, meaning that

---

**Algorithm 2** Sampling Training Waypoints in Space

---

    **Input:** Ground truth ego-agent track $s^i_{1:T_{\max}}$
            Range min/max distances $d_{\min}, d_{\max}$
            Maximum number of conditions $N$
    **Output:** Ordered list of conditions $C$

1:  $C \leftarrow \varnothing$
2:  $t_{\text{target}} \leftarrow 1$
3:  **do**
4:     Sample random distance $d_r \sim U(d_{\min}, d_{\max})$
5:     Find maximum $t_c \in \{t_{\text{target}}, \ldots, T_{\max}\}$ where $\|s^i_{t_c} - s^i_{t_{\text{target}}}\|_2 \leq d_r$
6:     $C \leftarrow C \cup \{s^i_{t_c}\}$
7:     $t_{\text{target}} \leftarrow t_c$
8:  **while** $\text{len}(C) < N$ and $t_{\text{target}} < T_{\max}$
9:  **return** $C$

---

not all agents are given a list of waypoints. Agents without waypoints are expected to react and behave realistically according to their learned human-like behavior priors.

Waypoints are provided to ITRA as part of the rendered ego-centric, ego-rotated birdview representation (Figure 1). This representation is well-suited for waypoint conditioning, as it allows for a natural placement of waypoints within the spatial context. Additionally, the limited field of view of the ego-centric representation enables the agent to act unconditionally until a waypoint enters its vicinity. Since each birdview is rendered from the perspective of each agent, we can naturally support unconditional generation for agents without a waypoint to reach by not including any waypoint circle in their raster representation.

**Sampling Training Waypoints:** The strategy for selecting training conditions is crucial. A straightforward method involves consistently using information from the last state of the training segment as the condition. For instance, this could mean relying solely on the position of the ego-agent at the final timestep $s^i_{1:T}$ of the training segment as the waypoint. We argue that this is not ideal since it implicitly introduces the concept of satisfying the condition exactly in $T$ timesteps. In Algorithm 2 we present a better sampling method for picking waypoints during training. Starting with an empty set of conditions $C$, the algorithm iteratively samples waypoints by selecting random distances within a defined range $[d_{\min}, d_{\max}]$. For each iteration, a random distance $d_r$ is sampled, and the algorithm searches for the farthest possible timestep $t_c$ such that the distance between the current waypoint and the target point is less than or equal to $d_r$. This found waypoint is then added to the list of conditions $C$. The process continues until the list contains a maximum number of conditions $N$ or the end of the ego trajectory $T_{\max}$ is reached. The algorithm ultimately returns the ordered list $C$ of sampled waypoints, which are used as training conditions.

### 3.4 Target Speed Conditioning

In many scenarios, controlling the aggressiveness of simulated driving behavior is essential for testing safety conditions. Driving aggressiveness can significantly affect safety outcomes, influencing the likelihood of collisions, near-misses, and the ability to navigate complex traffic situations. However, defining aggressiveness remains an open question in the literature, as it encompasses a wide range of behaviors and can have varying interpretations depending on the context (Danaf et al., 2015). For instance, aggressiveness may be reflected in rapid acceleration, sharp turns, or a tendency to follow other vehicles too closely. These behaviors can also differ depending on road conditions, traffic density, and even driving cultural factors.

Due to this complexity, directly modeling aggressiveness can be challenging. A practical, indirect method for controlling how aggressively a driver behaves is to condition their predicted actions on predefined target speed values. For instance, a lower target speed may lead to more cautious, conservative driving patterns, while a higher target speed could encourage more assertive or aggressive behaviors.

---

**Algorithm 3** Sampling Training Target Speeds in Time

---

**Input:** Ground truth ego-agent track $s^i_{1:T_{\max}}$
  Range min/max time increment $\Delta t_{\min}, \Delta t_{\max}$
  Maximum number of conditions $N$
**Output:** Ordered list of conditions $C$

1: $C \leftarrow \varnothing$
2: $t_{\text{target}} \leftarrow 1$
3: **do**
4:   Sample random time increment $\Delta t_r \sim U(\Delta t_{\min}, \Delta t_{\max})$
5:   Find maximum $t_c \in \{t_{\text{target}}, \ldots, t_{\text{target}} + \Delta t_r\}$ where $t_c \leq T_{\max}$
6:   $C \leftarrow C \cup \{s^i_{t_c}\}$
7:   $t_{\text{target}} \leftarrow t_c$
8: **while** $\text{len}(C) < N$ and $t_{\text{target}} < T_{\max}$
9: **return** $C$

---

To incorporate target speeds, we apply FiLM-like blocks (Perez et al., 2018) on the input of every intermediate layer of ITRA's encoder and decoder modules. Specifically, given a target speed $\bar{v}^i$ as condition and the recurrent state $h^i_t$ for agent $i$ at timestep $t$, we generate the scale and shift parameters for each layer $k$ as

$$\gamma^i_{t,k} = f_k(\bar{v}^i, h^i_t), \qquad \beta^i_{t,k} = h_k(\bar{v}^i, h^i_t). \tag{10}$$

These parameters are then used to perform conditional affine transformations of the input $\boldsymbol{x}^i_{t,k}$ of each layer by

$$\tilde{\boldsymbol{x}}^i_{t,k} = \gamma^i_{t,k} \boldsymbol{x}^i_{t,k} + \beta^i_{t,k}. \tag{11}$$

This process allows the model to adapt its feature representations based on the given target speed, effectively conditioning the driving actions on the desired speed profile. By design, we can allow for unconditional generation by assuming $\gamma^i_{t,k} = 1$ and $\beta^i_{t,k} = 0$ for agents that do not have a target speed condition. Target speed conditioning helps the model to capture the relationship between speed and other driving factors, such as road conditions and traffic density, leading to more realistic and robust driving behavior predictions. A target speed is regarded as *reached* when

$$|v^i_t - \bar{v}^i| \leq \epsilon_v, \tag{12}$$

where $v^i_t$ is the speed of the agent $i$ at timestep $t$ and $\epsilon_v$ is a small error coefficient.

**Sampling Training Target Speeds:** Similar to Section 3.3, we propose a strategy for sampling training target speeds that would allow for maintaining realistic driving behavior. Specifically, Algorithm 3 describes a process that generates an ordered list of training target speeds sampled in time contrary to Algorithm 2 that samples waypoints spatially.

## 4  Experiments

In this section, we begin by describing the experimental setup. We proceed by evaluating the performance of Control-ITRA through a series of experiments designed to measure the effectiveness of following waypoints and target speeds in various driving scenarios.

We train all our models on a large-scale self-driving dataset containing more than 1000 hours of traffic data collected from 19 countries worldwide. Drones were used to record continuous traffic trajectories from various kinds of intersections. Vehicles and pedestrians are represented by 2D bounding boxes that are automatically detected and tracked. Each location is annotated with a high-definition map representation capturing the road geometry and topology. In addition, traffic controls such as traffic lights, and stop and yield signs are annotated.

Table 1: Four-second ego-agent predictions given only initial state as observation. Conditions use information from the last ground truth ego state given at the ground truth segment. W and TS stand for the waypoint and target speed conditioning accordingly.

| Model | Cond. | ADE | minADE | FDE | minFDE | Miss Rate | MFD | Collision Rate | Waypoint Reach Rate | Target Speed Reach Rate |
|---|---|---|---|---|---|---|---|---|---|---|
| ITRA (Ścibior et al., 2021) | - | 0.93 | 0.44 | 2.46 | 1.07 | 0.14 | 6.59 | 0.01 | 0.73 | 0.83 |
| Control-ITRA (Last Timestep) | - | 0.95 | 0.46 | 2.52 | 1.10 | 0.14 | 6.51 | 0.01 | 0.75 | 0.81 |
| | W | 0.30 | 0.28 | 0.42 | 0.34 | 0.006 | 0.18 | 0.001 | 0.99 | 0.96 |
| | TS | 0.71 | 0.54 | 1.75 | 1.17 | 0.18 | 2.08 | 0.003 | 0.84 | 0.91 |
| | W/TS | 0.28 | 0.26 | 0.39 | 0.31 | 0.004 | 0.18 | 0.001 | 0.99 | 0.99 |
| Control-ITRA | - | 0.96 | 0.47 | 2.60 | 1.12 | 0.14 | 6.48 | 0.01 | 0.74 | 0.82 |
| | W | 0.63 | 0.32 | 1.32 | 0.54 | 0.07 | 3.73 | 0.005 | 0.80 | 0.89 |
| | TS | 0.75 | 0.41 | 1.89 | 0.93 | 0.11 | 4.35 | 0.003 | 0.80 | 0.88 |
| | W/TS | 0.50 | 0.28 | 0.96 | 0.44 | 0.04 | 2.15 | 0.001 | 0.89 | 0.95 |

Table 2: Eight-second ego-agent predictions given only initial state as observation. Conditions use information from the last ground truth ego state given at the ground truth segment. W and TS stand for the waypoint and target speed conditioning accordingly.

| Model | Cond. | ADE | minADE | FDE | minFDE | Miss Rate | MFD | Collision Rate | Waypoint Reach Rate | Target Speed Reach Rate |
|---|---|---|---|---|---|---|---|---|---|---|
| ITRA (Ścibior et al., 2021) | - | 3.21 | 1.44 | 8.63 | 3.44 | 0.45 | 20.29 | 0.04 | 0.61 | 0.78 |
| Control-ITRA (Last Timestep) | - | 3.14 | 1.82 | 8.58 | 4.47 | 0.50 | 14.53 | 0.04 | 0.61 | 0.79 |
| | W | 7.45 | 6.86 | 12.83 | 10.71 | 0.73 | 5.54 | 0.29 | 0.96 | 0.93 |
| | TS | 3.23 | 2.41 | 8.02 | 5.41 | 0.58 | 8.08 | 0.04 | 0.62 | 0.93 |
| | W/TS | 7.45 | 6.87 | 11.86 | 9.80 | 0.71 | 5.50 | 0.28 | 0.96 | 0.95 |
| Control-ITRA | - | 3.46 | 1.58 | 9.46 | 3.79 | 0.49 | 21.02 | 0.04 | 0.62 | 0.79 |
| | W | 2.18 | 1.11 | 3.61 | 1.28 | 0.44 | 8.55 | 0.03 | 0.77 | 0.90 |
| | TS | 2.87 | 1.50 | 6.93 | 3.24 | 0.42 | 6.93 | 0.03 | 0.69 | 0.93 |
| | W/TS | 2.06 | 1.21 | 3.21 | 1.43 | 0.41 | 5.48 | 0.02 | 0.84 | 0.94 |

All models are trained with 4-second segments with a simulation frequency of $10Hz$ which results in approximately 40 million segments usable for training. Only the first initial state is given as observation and the rest 39 timesteps are predicted. Similar to Ścibior et al. (2021), we used classmates-forcing during training where all states are replayed from the ground truth trajectory except for the states of the designated ego-agent. We set the introduced hyperparameters $R$ and $\epsilon_v$ to 2.0 and 1.0 accordingly.

## 4.1 Improving Performance By Following Ground Truth Conditions

We first test the ability of the proposed model to satisfy conditions in the same locations used for training. We use a validation set containing 1165 segments, each lasting four seconds. Our goal is to demonstrate that the conditional models can maintain realism while reaching the specified conditions. For this experiment, we provide only the initial state as an observation and generate subsequent timesteps. Similar to the training setting, we use classmates-forcing for the non-ego agents. We measure realism using multiple metrics. Specifically, we use the average displacement error (ADE) and the final displacement error (FDE) against the ground truth trajectory. For each validation case, we sample 6 predictions and additionally report the minimum ADE and FDE values of the six samples. Miss rate is also reported as an additional realism metric. A miss of an agent happens when at any point in the trajectory the distance from prediction to ground truth is higher than 2 meters. We also report the maximum final distance (MFD) metric (Ścibior et al., 2021) as

Table 3: Single-agent performance for waypoint conditioning using TorchDriveEnv. We generated 20 traffic initializations for each test location and sampled 4 predictions on the same initialization for all the tested models.

| Model | Condition | Collision Rate | Offroad Rate | Traffic Light Violation Rate | Avg. Number of Waypoints | Avg. Episode Length | Avg. Return |
|---|---|---|---|---|---|---|---|
| SAC | | 0.0 | 0.29 | 0.15 | 3.64 | 118.32 | 143.96 |
| PPO | Waypoints | 0.0 | 0.74 | 0.10 | 1.32 | 71.58 | 78.71 |
| TD3 | | 0.0 | 0.99 | 0.02 | 0.24 | 12.50 | 4.06 |
| A2C | | 0.0 | 0.98 | 0.01 | 0.19 | 16.12 | 6.31 |
| Control-ITRA | - | 0.0 | 0.08 | 0.21 | 2.06 | 170.79 | 297.98 |
| | Waypoints | 0.0 | 0.20 | 0.17 | 4.54 | 162.58 | 533.02 |

Table 4: Multi-agent performance for waypoint conditioning using TorchDriveEnv. We generated 20 traffic initializations for each test location and sampled 4 predictions on the same initialization for all the tested models.

| Model | Condition | Collision Rate | Offroad Rate | Traffic Light Violation Rate | Avg. Number of Waypoints | Avg. Episode Length | Avg. Return |
|---|---|---|---|---|---|---|---|
| SAC | | 0.34 | 0.27 | 0.14 | 2.34 | 108.93 | 105.17 |
| PPO | Waypoints | 0.24 | 0.62 | 0.15 | 1.15 | 59.42 | 51.24 |
| TD3 | | 0.11 | 0.91 | 0.01 | 0.20 | 10.68 | 4.89 |
| A2C | | 0.14 | 0.84 | 0.02 | 0.28 | 13.22 | 7.11 |
| Control-ITRA | - | 0.21 | 0.11 | 0.10 | 1.25 | 142.47 | 182.46 |
| | Waypoints | 0.11 | 0.02 | 0.09 | 2.75 | 167.45 | 317.53 |

a measurement of diversity in the sampled predictions. It is important for the conditional driving model to satisfy conditions while not yielding additional infractions. We report collision rate to showcase the ability of the model to not drive recklessly for the sake of condition reachability. Finally, we state the rate of reaching both the waypoint and target speed conditions.

In Table 1 we compare three different models. As a baseline, we trained a standard unconditional ITRA model as described in Ścibior et al. (2021) and reported the results on all metrics. Although this model does not support waypoint or target speed conditioning, we still report the average rate of reaching the last-timestep conditions as previously defined. In Section 3.2, we mentioned that a rather straightforward way for picking training conditions is to always use the information from the last timestep of the training segment. We compare this strategy (referred to as *last timestep*) against our proposed way of sampling training conditions. For every conditional model, we test their unconditional capabilities as well as their ability to satisfy either condition or both at the same time. As expected, all conditional models achieve higher condition-satisfaction rates than either the baseline ITRA model or the conditional models when tested without providing conditions. Notably, models trained with the *last timestep* sampling strategy perform better than those trained with our proposed sampling scheme. This occurs because training with waypoints always positioned at the fourth second in the ground-truth trajectory implicitly encourages the model to reach waypoints precisely at four seconds, which improves performance on this specific experiment by reinforcing ground-truth trajectory adherence.

However, as shown in Table 2, when we test the same models on eight-second predictions given only the initial state as observation, the performance of the model trained with the *last timestep* strategy significantly declines. Although it still satisfies the conditions at a higher rate, its collision rate becomes unacceptable, and its realism metrics suffer. This degradation occurs because the model rushes to reach the waypoint sampled from the eight-second timestep at exactly four seconds, compromising realistic driving behavior.

Table 5: Results on target speed conditioning using the unseen locations from TorchDriveEnv. Episodes run for 20 seconds using the five test locations. For each location and target speed, we used 30 different traffic initializations and sampled 4 generated trajectory rollouts from the stochastic Control-ITRA model.

| Target Speed (km/h) | Traffic | Condition Given | Collision Rate | Offroad Rate | Traffic Light Violation Rate | Target Speed Hit Percentage |
|---|---|---|---|---|---|---|
| 0 | Single-Agent | No | - | 0.08 | 0.21 | 29.8% |
| | | Yes | - | 0.14 | 0.11 | 84.3% |
| | Multi-Agent | No | 0.21 | 0.11 | 0.10 | 39.6% |
| | | Yes | 0.18 | 0.16 | 0.06 | 76.3% |
| 20 | Single-Agent | No | - | 0.09 | 0.21 | 71.5% |
| | | Yes | - | 0.10 | 0.17 | 79.8% |
| | Multi-Agent | No | 0.23 | 0.10 | 0.10 | 68.0% |
| | | Yes | 0.23 | 0.12 | 0.12 | 72.6% |
| 35 | Single-Agent | No | - | 0.10 | 0.21 | 36.8% |
| | | Yes | - | 0.09 | 0.31 | 65.5% |
| | Multi-Agent | No | 0.21 | 0.11 | 0.11 | 28.5% |
| | | Yes | 0.35 | 0.09 | 0.19 | 46.8% |
| 55 | Single-Agent | No | - | 0.10 | 0.22 | 6.0% |
| | | Yes | - | 0.07 | 0.45 | 32.0% |
| | Multi-Agent | No | 0.22 | 0.09 | 0.13 | 4.0% |
| | | Yes | 0.40 | 0.08 | 0.23 | 21.8% |
| 70 | Single-Agent | No | - | 0.09 | 0.22 | 1.0% |
| | | Yes | - | 0.05 | 0.41 | 5.0% |
| | Multi-Agent | No | 0.21 | 0.12 | 0.11 | 0.3% |
| | | Yes | 0.37 | 0.09 | 0.23 | 2.5% |
| 90 | Single-Agent | No | - | 0.11 | 0.23 | 0.0% |
| | | Yes | - | 0.03 | 0.37 | 0.6% |
| | Multi-Agent | No | 0.22 | 0.09 | 0.10 | 0.0% |
| | | Yes | 0.40 | 0.11 | 0.19 | 0.6% |
| 110 | Single-Agent | No | - | 0.08 | 0.19 | 0.0% |
| | | Yes | - | 0.05 | 0.35 | 0.3% |
| | Multi-Agent | No | 0.22 | 0.11 | 0.09 | 0.0% |
| | | Yes | 0.41 | 0.12 | 0.17 | 0.1% |

## 4.2 Testing Out-of-domain Performance

We also evaluate the model's performance in new, unseen locations to ensure that it generalizes well across various scenarios while maintaining both condition satisfaction and good driving behavior. For this testing, we leverage TorchDriveEnv (Lavington et al., 2024), a reinforcement learning environment with simulated traffic driven by a human-like expert policy model. TorchDriveEnv utilizes locations from the CARLA simulator (Dosovitskiy et al., 2017) and enables the control of a designated ego agent, while the rest of the non-player characters (NPCs) are driven to create realistic traffic. In TorchDriveEnv, as with ITRA-based models, the action space is continuous and defined by steering and acceleration, and observations are provided as 2D egocentric birdview rasterizations. The reward function is given by

$$r = \alpha_1 r_{\mathrm{movement}} + \alpha_2 r_{\mathrm{waypoint}} - \beta_1 r_{\mathrm{smoothness}}, \tag{13}$$

where $\alpha_1$, $\alpha_2$, and $\beta_1$ are hyperparameters. We adopt the default configuration from the released benchmark codebase[1].

The environment includes five distinct validation scenarios: Parked-Car, Three-Way, Chicken, Roundabout, and Traffic-Lights. Each scenario is designed to test the model's capability to navigate specific challenging

---

[1] https://github.com/inverted-ai/torchdriveenv

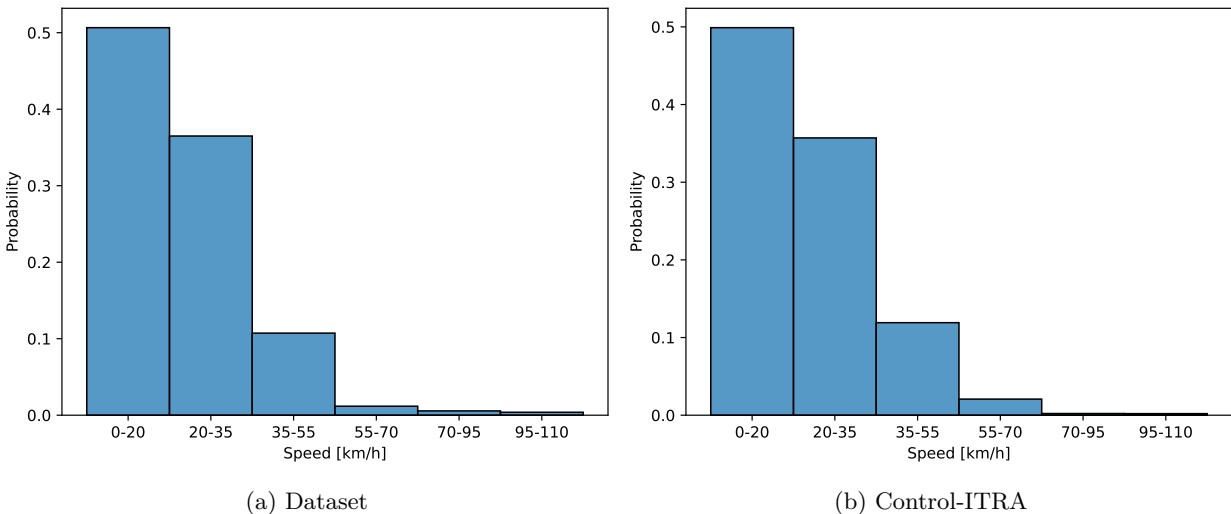

Figure 2: The distribution of speed values in the collected human-traffic training dataset compared to the learned speed distribution of Control-ITRA.

situations. For each scenario, a designated ego agent is assigned, while the remaining traffic agents are randomly initialized and reactively simulated using a commercial simulation service. The ego agent is given a sequence of waypoints, and the simulation halts if any infraction (e.g., collisions, off-road driving, or traffic light violations) involving the ego agent occurs. We evaluate our approach in both single-agent (without other traffic agents) and multi-agent (with other traffic agents) settings. As a baseline, we report the performance of four standard reinforcement learning algorithms—SAC (Haarnoja et al., 2018), PPO (Schulman et al., 2017), TD3 (Fujimoto et al., 2018), and A2C (Mnih et al., 2016)—trained in the same environment following the setup in Lavington et al. (2024). For each method, we report the average cumulative return (as defined in Equation (13)), average episode length and the average number of waypoints reached. Additionally, we measure the infraction rates for collisions, off-road incidents, and traffic light violations.

As shown in Table 3, in the single-agent setting, Control-ITRA outperforms all baseline RL methods, achieving a higher average number of waypoints reached and a higher cumulative return. The smoothness penalty in the reward function causes RL baseline methods to suffer from excessive jerk movements, which contributes to their lower average returns despite reaching comparable waypoint counts. In contrast, Control-ITRA, being a data-driven approach trained on imitating human-collected traffic data, produces notably smoother trajectories. Additionally, running Control-ITRA unconditionally results in fewer waypoints reached, highlighting the model's effectiveness in following waypoints when conditioned to do so.

In the multi-agent setting (Table 4), Control-ITRA also achieves a higher average return and reaches more waypoints compared to the RL baseline methods. The driving behavior is smoother (as implied by the reward function), resulting in longer episodes with significantly lower infraction rates in both conditional and unconditional prediction modes.

As of the time of writing, TorchDriveEnv does not include standard test cases to assess target speed conditioning. Therefore, we evaluate the model's ability to follow target speeds in new, unseen locations by testing on the same five scenarios from TorchDriveEnv, while conditioning on seven target speeds. We conduct this experiment in both single-agent and multi-agent settings, with results presented in Table 5. The model satisfies the target speed condition at a significantly higher rate, particularly for lower speeds, compared to unconditional predictions, with minimal compromise in infraction rates. However, as target speeds increase, the model shows a tendency toward higher collision rates. This is expected since target speed functions as an implicit control for aggressiveness. Additionally, we observe that hit percentages for high speeds decrease, which can be attributed to three factors. First, TorchDriveEnv test locations feature single-lane roads that are not conducive to safely reaching high speeds. Initial states from TorchDriveEnv contain pre-defined initial speeds that are given as input to the model. It is highly unlikely that these initial speeds are initialized

in a way that would allow agents to reach high target speeds. Second, episodes terminate after 20 seconds, which may limit the model's ability to accelerate to high speeds realistically. Finally, as shown in Figure 2, the dataset used to train our model contains few instances of high-speed values, limiting the model's training opportunities for high target speed conditioning. In the same figure, we can see that Control-ITRA very closely imitates the speed distribution of the dataset.

## 5    Conclusion

In this paper, we highlighted the importance of controlling driving behavior through waypoint setting and indirectly modulating behavior aggressiveness by conditioning on target speeds. We extended the ITRA driving behavior model to enable partial conditioning of agents in the scene to follow waypoints, target speeds, or both. We proposed Control-ITRA, a training scheme that allows the model to adhere to these control conditions while maintaining realistic, human-like driving behavior.

Our experiments demonstrated that in locations where traffic data is available, the conditional model effectively follows waypoints and target speeds without compromising behavioral realism. Additionally, we validated the method in novel, unseen locations, showing that it can satisfy the given conditions without increasing infraction rates. These controllable models offer the potential for augmenting current driving simulations to create complex and challenging scenarios. Future work could explore conditioning on more abstract control forms, such as natural language commands or driver intentions.

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
