# OpenReview forum: "Control-ITRA: Controlling the Behavior of a Driving Model"
_TMLR — Rejected by TMLR_

### Review · Reviewer_bsPZ · 2024-11-24

**Summary Of Contributions:**

This paper proposes a method called control-ITRA for controllable driving simulation based on the variational autoencoder/variational RNN framework. This family of models learns human driving trajectories by inferring latent controls $a$ and states $z$. In order to generate specific trajectories upon queries, the authors propose to condition the model on 1) waypoints that are added into the birds eye view driving scene one by one and 2) target speed that are fed into the encoder and decoder layers. A data sampling based conditional training strategy is used which "bootstrap" as set of conditioning variables based on feasibility. Experiments were conducted to verify the model's condition satisfaction ability and realism, showing superior performance over baselines.

**Audience:**

Yes

**Broader Impact Concerns:**

There is no ethical implications of this work that needs to be addressed.

**Claims And Evidence:**

Yes

**Requested Changes:**

**On the background section**
* Repeating the previous point, I think the authors should conduct a slightly more comprehensive review on state of the art in controllable driving simulation.

**On the method section**
* I think it's helpful to add a short summary at the top of the method section to prepare the reader. In the first pass I was confused about how conditioning is introduced to the model until the final part of this section (3.4).
* Following the previous point, it might be easier for the authors to move parts of section 3.4 immediately after 3.1 and introduce some notation for the conditioning strategy, e.g., $p(s_{1:T}|C)$. It may then become easy for the authors to explain that $C$ is never explicitly represented in the dataset but can be curated using the proposed sampling strategy.
* Since this type of notation was never formally introduced, it was not clear how the model is supposed to behave without conditioning. The authors introduced notation $p(s_{t}|s_{1:t-1}, \emptyset)$ in algo 1. Does it mean conditioning on some padding values?
* Eq 1 modeling choice: the choice of using $p(z_{t})$ as a fully factorized latent variable distribution that could in principle change to some random values at every time step is very curious. I understand the authors inherited this model from the previous version ITRA, but it might be helpful to explain it to curious readers. Since the only distribution that conditions on this is $p(a|b, z)$, which is essentially the control policy, it's tempting to interpret this as modeling latent driver attributes. Then one would expect some consistency of driver attributes across time steps rather than being independent.

**On the experiment section**
* When the authors test ADE in conditioned scenarios, which ground truth trajectories are being compared to? Because if arbitrary conditioning variables are selected, then the dataset trajectories are no longer ground truth for those scenarios. My guess is the same conditioning variable generation strategy from training was used for selected test trajectories. It was hard for me to spot where this was stated in the paper.
* What exactly makes the test scenarios in section 4.2 out-of-domain? Is it because it uses CARLA maps which are different from the training ones? It was hard for me to confirm this since the beginning of section 4.2 put a lot of emphases on describing the TorchDriveEnv.
* Why are the RL agents much worse than the proposed model, given they have the same architecture which receive the same type of inputs and generate the same type of outputs?
* Minor: I don't think the authors ever explained what miss rate and MFD mean. I got the definition of MFD as max final distance from ITRA paper.

**Strengths And Weaknesses:**

**Strength**
* This paper tackles an important problem with a well-motivated methodology.
* The experiments are extensive and show clear success of achieving desired outcomes, which is controllability and realism, and enhancement in performance over baselines.

**Weakness**
* The organization and writing can be improved. I think certain details should be added or highlighted to make it easier for readers to understand the work. I discuss more details below.
* I suspect the background on controllable driving simulation is limited, which makes it difficult for readers to have a holistic understanding of the state of the art and where the propose method stands. I found the following work through a quick Google search which seem to directly tackle this area but using very different methodology from the current work:

[1] [CtRL-Sim: Reactive and Controllable Driving Agents with Offline Reinforcement Learning, 2024](https://arxiv.org/abs/2403.19918)

[2] [SceneDiffuser: Efficient and Controllable Driving Simulation Initialization and Rollout (2024)](https://openreview.net/forum?id=a4qT29Levh&referrer=%5Bthe%20profile%20of%20Chiyu%20Max%20Jiang%5D(%2Fprofile%3Fid%3D~Chiyu_Max_Jiang1))

---

> ### Author Response · Authors · 2024-12-18
> **Response to Reviewer bsPZ (Part 1)**
>
> We sincerely thank the reviewer for their encouraging and helpful feedback that will help improve our paper.
>
> - **Regarding the comment “I think the authors should conduct a slightly more comprehensive review on state of the art in controllable driving simulation”:**
>
> We have revised the paper and expanded the related work section to include a more comprehensive review of existing methods in controllable driving simulation. We kindly invite the reviewer to review these updates in the revised manuscript.
>
> - **Regarding the method section:**
>
> Following the reviewer's recommendations, we revised the methodology section. Specifically, we moved Section 3.4 immediately after 3.1 and reorganized the placement of the algorithmic blocks.
>
> - **Regarding the question “Since this type of notation was never formally introduced, it was not clear how the model is supposed to behave without conditioning. The authors introduced notation $p(s_{t}|s_{1:t-1}, \emptyset)$ in algo 1. Does it mean conditioning on some padding values?”:**
>
> We designed the introduction of conditions such that passing no condition for a specific agent defaults to the same unconditional execution as the base model. Specifically, in the ego-rotated and ego-centric birdview representation - the main representation used by agents to perceive the environment - we include an additional input specifying waypoint positions per agent and a waypoint mask that indicates whether the waypoint is valid for that agent. If the mask is set to true, an additional waypoint mesh is rendered in the agent-specific birdview. Otherwise, the agent perceives the same representation as the base unconditional model, without any waypoint included.
>
> Similarly, for target speed conditioning, we apply conditional transformation layers between the base network’s existing layers. When the target speed conditioning mask is set to false, we assume an identity transformation, which leaves the intermediate network representations unchanged. This ensures that agents without conditioning behave identically to the base unconditional model.
>
> - **Regarding the comment “Eq 1 modeling choice: the choice of using as a fully factorized latent variable distribution that could in principle change to some random values at every time step is very curious. I understand the authors inherited this model from the previous version ITRA, but it might be helpful to explain it to curious readers. Since the only distribution that conditions on this is , which is essentially the control policy, it's tempting to interpret this as modeling latent driver attributes. Then one would expect some consistency of driver attributes across time steps rather than being independent.”:**
>
> We appreciate the reviewer's thoughtful observation about the temporal independence of the latent variables $z_t$. While the current formulation does indeed allow for potentially different random values at each timestep, there are several important considerations that provide implicit temporal consistency:
>
> - The latent variables $z_t$ influence the agent's actions through the recurrent hidden states $h_t$, which maintain temporal continuity through the RNN dynamics. Since $h_t$ depends on the previous hidden state $h_{t-1}$, actions $a_t$, and current latent $z_t$, the model can learn to integrate the latent variables in a temporally consistent way.
> - During training, the model learns to utilize the latent space in a way that produces coherent trajectories, effectively enforcing behavioral consistency through the optimization process rather than through explicit constraints on $z_t$.
> - The reviewer's interpretation of $z_t$ as modeling driver attributes is interesting and aligns with how the model often ends up using these variables in practice. While the variables are technically independent across timesteps, the recurrent structure allows the model to learn stable mappings between regions of the latent space and consistent driving styles.
>
> That said, we acknowledge that explicitly modeling temporal dependencies in the latent space (e.g., through a temporal prior $p(z_t|z_{t-1})$) could potentially lead to more interpretable representations of driver attributes. This is an interesting direction for future work that could help bridge the gap between the current flexible but independent formulation and more structured approaches to modeling driver characteristics.

---

> ### Author Response · Authors · 2024-12-18
> **Response to Reviewer bsPZ (Part 2)**
>
> - **Regarding the question “When the authors test ADE in conditioned scenarios, which ground truth trajectories are being compared to? Because if arbitrary conditioning variables are selected, then the dataset trajectories are no longer ground truth for those scenarios. My guess is the same conditioning variable generation strategy from training was used for selected test trajectories. It was hard for me to spot where this was stated in the paper.”**
>
> For the experiments in Section 4.1, we use a collection of validation ground truth data that is not used during training. While the location selection overlaps with locations in the training set, all scenarios tested are unseen. Specifically, we initialize each scene with the first state of the ground truth segment and use the last ground truth state of the ego agent as the waypoint and target speed condition. This applies to both the 4-second and 8-second trajectory segments.
>
> During testing, all non-ego agents are replayed using their ground truth trajectories (class-mates forcing), while the ego agent samples latents from the prior distribution during the generation process. This setup ensures that the conditions provided to the ego agent are extracted directly from the known ground truth trajectory.
>
> The purpose of this experiment is to demonstrate that the model, when provided with conditions derived from ground truth, can generate predicted behaviors that are closer to the ground truth trajectory than those generated unconditionally. We hope this clarifies any confusion, and we are happy to address any further questions.
>
> - **Regarding the question “What exactly makes the test scenarios in section 4.2 out-of-domain? Is it because it uses CARLA maps which are different from the training ones? It was hard for me to confirm this since the beginning of section 4.2 put a lot of emphases on describing the TorchDriveEnv.”:**
>
> You are correct that we consider out-of-domain scenarios to be those set in locations completely unseen during training. In this case, the CARLA locations qualify as out-of-domain because they are entirely synthetic and do not include recorded human-generated traffic scenarios.
>
> - **Regarding question “Why are the RL agents much worse than the proposed model, given they have the same architecture which receive the same type of inputs and generate the same type of outputs?”:**
>
> The TorchDriveEnv paper [1] describes an experimental setup involving basic baselines designed primarily to demonstrate their proposed driving environment rather than to optimize performance. For our baseline results, we followed their exact setup. It is important to note that their configuration is not tuned for high performance but instead serves as a proof of concept for applying four RL methods in that environment.
>
> The RL methods tested are trained without any focus on realism, solely aiming to maximize the expected reward as defined by the TorchDriveEnv framework. While SAC and PPO perform better than the other two methods, they exhibit excessive jerkiness, resulting in lower average returns compared to our proposed models, which are explicitly trained to imitate human-generated trajectories.
>
> The authors of TorchDriveEnv also discuss the challenges of training RL methods in a multi-agent setting, which aligns with our observations. In our work, we do not use this environment for training; instead, we use it exclusively for testing our models trained with real-world data.
>
> To the best of our knowledge, we are the first to conduct experiments using this driving simulation environment, making direct comparisons with other methods challenging.
>
> - **Regarding comment “Minor: I don't think the authors ever explained what miss rate and MFD mean. I got the definition of MFD as max final distance from ITRA paper.”:**
>
> We apologize for this oversight. In the revised version of the paper, we have added a description of both the miss rate and MFD metrics to clarify their definitions.
>
> [1] [TorchDriveEnv: A Reinforcement Learning Benchmark for Autonomous Driving with Reactive, Realistic, and Diverse Non-Playable Characters](https://arxiv.org/abs/2405.04491)

---

> > ### Comment · Reviewer_bsPZ · 2024-12-18
> >
> > Thank the authors for the clarifications and updates. A very minor suggestion for future rebuttals: highlighting changes in a different color in the manuscript will make things much easier for reviewers.
> >
> > A follow up question on conditioned scenarios: so in principle there's no ground truth or straightforward way to make comparisons when the conditioning waypoints and target speed differ from the dataset? I suspect this is a general problem with controllable generation and maybe the community will address this together in the future.
> >
> > On the RL results, the reason why it's curious is that the RL agents are specifically trained to optimize returns, whereas C-ITRA is not. So it's a bit surprising to see C-ITRA achieves higher rewards, because generally we would expect the RL agents to do very well, often over-optimizing the rewards at the expense of realism. I'm guessing this is related to the difficulty of optimization/exploration for RL.

---

> > > ### Author Response · Authors · 2024-12-20
> > > **Re: Official Comment by Reviewer bsPZ**
> > >
> > > We would like to thank the reviewer for taking the time to review our revised manuscript and for providing a response to our follow-up.
> > >
> > > - **Regarding your question “so in principle there's no ground truth or straightforward way to make comparisons when the conditioning waypoints and target speed differ from the dataset? I suspect this is a general problem with controllable generation and maybe the community will address this together in the future.”:**
> > >
> > > In general, under our problem setting, there are two broad aspects to evaluate: (1) How “human-like” or “realistic” the generated driving behavior is, and (2) How often the generated behavior satisfies a given condition. Our aim is to simulate driving behavior that is both realistic and adheres to reasonable, “realistic” conditions. Depending on the definition of the condition, measuring whether it is satisfied can be nontrivial or even infeasible. In our case, we quantify waypoint and target speed satisfaction as defined in Sections 3.3 and 3.4.
> > >
> > > In the literature, evaluating the realism of traffic simulations remains largely an open problem, particularly in the absence of human demonstrations. When human data is available, standard metrics for measuring realism include (min)ADE, (min)FDE, and miss rate. In novel settings where such data is unavailable, proxy metrics are often used to assess general characteristics of the generated driving behavior, such as collision rate and offroad rate.
> > >
> > > Since our goal is to demonstrate that we can preserve the imitated “human-like” driving behavior while achieving high rates of condition satisfaction, the experiments in Section 4.1 are designed to approximately test this claim. Naturally, there are multiple realistic ways to achieve a given condition, especially when predicting longer horizons. Because we lack access to the true distribution of human-like actions, testing the true realism of all conditional (and unconditional) predicted behaviors is challenging.
> > >
> > > For Section 4.2, we rely on infraction metrics to assess the reasonableness of the generated trajectories.

---

> > > > ### Comment · Reviewer_bsPZ · 2024-12-20
> > > >
> > > > Thank the authors again for the clarification. The concern is very reasonable and the proxy metrics do follow current best practices as far as I know.

---

### Review · Reviewer_eN7B · 2024-12-04

**Summary Of Contributions:**

the paper propose a variation of ITRA driver models, where additional to-be-accomplished conditions can be manually specified and fed into the models, such as waypoints (a-temporal, i.e., just a rough path that should be followed, in constrast to temporal trajectory matching), or target speed.

the conditions are incoporated as follows:
- the ITRA model is modified, such that besides the usual input, it takes the additional conditions, e.g., by incorporating the target waypoints into the birds eye view image that ITRA already uses as state/scene input.
- the training happens by randomly sampling samples with and without conditions, such that the model learns to cover both cases. the waypoints conditions are taken from the given data in a self-supervised way, in a specific way that the model at each step gets a list of reachable future waypoints, instead of just getting the last waypoint (to avoid being confined to reaching waypoints perfectly according to ground truth timing).

the proposed method is compared to various baslines within distribution (imitation), as well as in an OOD simulation environment against RL baselines.

**Audience:**

Yes

**Broader Impact Concerns:**

-

**Claims And Evidence:**

Yes

**Requested Changes:**

either MixSim should be included in the experimental comparison, or a good, explicit reason should be given why it is not compared against.

additionally, some minor improvements are necessary w.r.t. the writing of the notations/model parts, etc., as stated above.

update after rebuttal
---

changed "Claims And Evidence" to "Yes", see comments.

**Strengths And Weaknesses:**

strengths/mixed points
---

the paper approaches an important problem. as the authors state, manual controlability of learned driver models is particularly important to cover long-tail safety-critical events that, by definition, are too rare in most of real world recorded human driving data. the proposed method is on the heuristic side of approaches, as often in this field. i would have hoped for a bit more principled way, maybe using some sort of variational approximations to get a conditional model out of the original ITRA model without having to expliticitly train in that way, which would mean one single model is ad-hoc transferable to the new, conditional task. but nonetheless the porposed method is a valid way.

the experiments show some interesting insights:
- it is interesting to see how the last-waypoint-conditioning works "in domain" in table 1, but not "out domain" (table 2) when giving 8-second instead of 4-second goals, as discussed in sec4.1. however, the comparison seems a bit unfair, because this is simply not what the last-waypoint-conditioning was trained for, right?
- as a broader comment beyond the specific task, it is interesting to see that Control-ITRA also seems to perform well OOD, because this is a huge issue for all existing IL-based methods in AD.

overall, notation and model assumptions are properly introduced to some extent, but few things are not explicitly introduced or derived. i'm not asking to make everyithing fully explicit, but a bit more detail here and there would not harm:
- like the mu_theta of the policy, and specifically, what is the scene perception backbone applied to the birds eye view grid, i imagine a CNN? additionally, a bit of derivative math seems to be missing for how to get from the base conditionals to the conditionals in eq (7) but again this is not central to the present paper.
- minor thing, just out of curiousity: given that some of the conditional distributions are dirac distributions, so they don't have a Lebesgue density, so what kind of integral is eq (1) if interpreting it mathematically rigorously (which is not central to the paper).
- what i don't get is where the h in eq 7 are specified. i imagine this somehow follows form those mentioned variational RNNs, but also here a brief note for readers which may be familiar with VAE basics, but not variational RNNs would be interesting.

minor downside: the birds eye view is a very high dimensional signal to incorporate the waypoint, and i would be afraid that this is less robust than incorporating waypoints via scalars or the like.

weaknesses (in particular comparison to important related work missing) and unclearities
---

in the experiments, some minor questions occured to me. they are not necessarily a weakness, but maybe something to be clarified:
- why is Control-ITRA with W and TS worse than with just W in table 2? i guess it can easily be explained by the higher model complexity, however, from an application side, i think it is rather undesiarable if, the more conditions you give to the method, the worse it performs in reaching the condition.
- in sec4.2, how many episodes are used to train those RL-based baselines? because with enough episodes one would expect them to do better, and they may also generalize better due to some borader exploration during training. but this is just a guess.

in my view, the biggest weakness of the paper is that *at least* one important method -- MixSim [1] -- is missing in related work as well experimental comparisons. MixSim also introduces controllability of scenarios to learned traffic simulations. note that there is quite some overlap in the problem formulation, in particulat what Control-ITRA refers to as waypoints, is, as far as i see, more or less the same as what MixSim refers to as the nodes of a route (just a different representation, and bit a higher level of granularity). if i'm not mistaken, MixSim has exactly that ability that also Control-ITRA claims, which is that waypoints (route nodes) do not have to be reached as specific time steps in the trajectory, but are a-temporal paths.

i.e., the baselines in the experimental comparison seem to be just simple baselines, and do not cover actual competitors on the very task under consideration such as MixSim.

note that by now there may be more work like MixSim, which i am not aware of myself, and that should be checked and potentially included.

[1] Simon Suo,*  Kelvin Wong,*  Justin Xu,  James Tu,   Alexander Cui,  Sergio Casas,  Raquel Urtasun: MixSim: A Hierarchical Framework for Mixed Reality Traffic Simulation

conclusion
---

i have a hard time giving a conclusion of my review. i could not say that the claims made in the paper are explicitly unsupported by the evidence in the paper. however, including closely related work/baselines should probably be part of a good claim, or it should at least be argued why not including such baselines.

in this sense, for now, i'm giving a "No" for "Claims And Evidence", but i'm open to arguments why MixSim (or other related work in this direction) should not be included in the present work.

---

> ### Author Response · Authors · 2024-12-18
> **Response to Reviewer eN7B (Part 1)**
>
> We sincerely thank the reviewer for their encouraging and helpful feedback that will help improve our paper.
>
> - **Regarding the question “What is the scene perception backbone applied to the birds eye view grid, i imagine a CNN?”:**
>
> Similar to the base ITRA model, each agent is modeled with a two-layer recurrent neural network using the gated recurrent unit (GRU) with a convolutional neural network (CNN) encoder for processing the birdview image. The remaining components are fully connected.
>
> - **Regarding the comment “Given that some of the conditional distributions are dirac distributions, so they don't have a Lebesgue density, so what kind of integral is eq (1) if interpreting it mathematically rigorously (which is not central to the paper)”:**
>
> Thank you for pointing this out. You are correct that some of the conditional distributions in our model are Dirac delta distributions, which are not proper probability density functions with respect to Lebesgue measure. More rigorously, equation (1) should be interpreted as an integral with respect to the appropriate product measure. Specifically, the measure space should be constructed as a product measure where:
>
> - The components involving Gaussian distributions (p(z_t^i) and p_θ(a_t^i|...)) are with respect to Lebesgue measure.
> - The components involving Dirac deltas (the render function, RNN state transitions, and kinematic model) are with respect to counting measure.
>
> As a result, the integral in equation (1) is not a standard Lebesgue integral but a constrained integral, where these delta distributions effectively reduce the integration domain to the submanifold defined by the deterministic constraints. This means the integral should technically be written using the language of measure theory, where the joint distribution is defined through the Radon-Nikodym derivatives with respect to this product measure. However, we chose to maintain the more familiar density notation for clarity and accessibility, as is common practice in the machine learning literature when dealing with mixed continuous-discrete systems.
>
> The practical implementation and results of our model remain unchanged by this technical consideration, as the sampling process naturally handles both the continuous and deterministic components correctly.
>
> - **Regarding the comment “What i don't get is where the h in eq 7 are specified. i imagine this somehow follows form those mentioned variational RNNs, but also here a brief note for readers which may be familiar with VAE basics, but not variational RNNs would be interesting”:**
>
> The h in Equation 7 represents the recurrent states generated by Equation 5. To keep the notation in Equation 7 concise, we did not explicitly include this term in the loss formulation. However, we have added a follow-up explanation in the revised manuscript to clarify the origin of h for readers who may be familiar with VAE basics but are less familiar with variational RNNs.
>
> - **Regarding the comment “The birds eye view is a very high dimensional signal to incorporate the waypoint, and i would be afraid that this is less robust than incorporating waypoints via scalars or the like”:**
>
> In our preliminary experiments, we observed that adding waypoints directly to the rendered birdview provided superior results compared to using scalar-based representations. Furthermore, we believe this approach offers greater flexibility, enabling more advanced conditioning methods.
>
> For instance, this representation can be seamlessly extended to display multiple waypoints simultaneously without increasing the neural network’s computational speed or memory requirements. This capability can allow the model to make informed choices between different waypoints (e.g., selecting the first or second exit on a roundabout while avoiding the third). Additionally, it facilitates enhancements such as color-coding or dynamically sizing the waypoints to signal more complex driving maneuvers.
>
> While these advanced applications are beyond the scope of the current submission, we consider them promising directions for future work.

---

> ### Author Response · Authors · 2024-12-18
> **Response to Reviewer eN7B (Part 2)**
>
> - **Regarding the question “Why is Control-ITRA with W and TS worse than with just W in table 2? i guess it can easily be explained by the higher model complexity, however, from an application side, i think it is rather undesiarable if, the more conditions you give to the method, the worse it performs in reaching the condition.”:**
>
> In Table 2, Control-ITRA conditioned on both waypoints and target speed (W/TS) achieves a higher condition reach rate compared to Control-ITRA with waypoint conditioning alone (W). Additionally, on average, all six generated samples in the W/TS setting are closer to the ground truth in terms of ADE and FDE.
>
> The slightly higher values of minADE and minFDE in the W/TS case can be attributed to the inherent randomness of the model. Sampling more trajectories would likely result in lower values for these metrics. Furthermore, because this is a longer horizon task (8-second segments), there are naturally more ways to reach a given waypoint the further the agent drives, which introduces variability.
>
> It is important to highlight that a waypoint is considered "reached" when the agent is within a radius R (set to 2 meters in our experiments). In contrast, FDE measures the precise distance between the final predicted state and the final state of the ground truth trajectory. This means agents conditioned on W/TS may reach waypoints slightly earlier or deviate marginally in their final position relative to the ground truth.
>
> The primary purpose of this experiment is to demonstrate that predictions conditioned on both waypoints and target speed are generally closer to the ground truth compared to those generated unconditionally. While variability exists in specific metrics, the results confirm that adding conditions enhances the alignment of generated trajectories with real-world data.
>
> - **Regarding the question “In sec4.2, how many episodes are used to train those RL-based baselines? because with enough episodes one would expect them to do better, and they may also generalize better due to some borader exploration during training. but this is just a guess.”:**
>
> Similar to the TorchDriveEnv paper [1], we trained these models for 1 million environment steps. Each episode terminates either when an infraction is detected (collision or going off-road) or after 200 steps have elapsed. Episodes are randomly initialized through the environment's internal process, making it unlikely to encounter identical episodes unless a specific random seed is set during initialization. That being said there is no set number of episodes used during training.
>
> - **Regarding MixSim:**
>
> We would like to thank the reviewer for bringing this work to our attention, which we have added as prior art in the related work section. We believe this is definitely relevant work that readers should be aware of, but we do not see it as something we need to directly compare with. Our reasoning is:
>
> - As correctly pointed out by the reviewer, the definition of a goal in MixSim differs from ours. In their work, goals are defined as routes that are directed paths of lane segments composed of sequences of roadgraph nodes. In our case, since our definition of waypoints is independent of any road definition, we allow placing waypoints anywhere on the map, which can potentially be used for any kind of waypoint conditioning (i.e., to execute specific maneuvers based on the given waypoints).
> - Based on our understanding, for multi-agent predictions, MixSim requires all agents to have a goal condition as input. Their definition of mixed reality relies on the fact that each agent can take different route conditions but still constrains generation such that all agents must have a condition given. The majority of their paper (Section 3.3) discusses interesting ways of generating such conditions at test time for simulating desired scenarios.
> - Sections 3.1 and 3.2 of the MixSim paper do not describe the process of extracting the goal conditions used during the training of their model. We would argue that if the authors extract such conditions directly from the ground truth data, our proposed Control-ITRA sampling scheme for training conditions would be beneficial for MixSim.
>
> [1] [TorchDriveEnv: A Reinforcement Learning Benchmark for Autonomous Driving with Reactive, Realistic, and Diverse Non-Playable Characters](https://arxiv.org/abs/2405.04491)

---

> > ### Comment · Reviewer_eN7B · 2024-12-19
> >
> > thanks to the authors for the responses and explanations to my various points.
> >
> > regarding my main concern on inclusion of relevant related work, i cannot fully follow the given argument why MixSim is not included in the experimental comparison. i see that there might be subtle differences in the setups, but without such external baseline comparisons (against MixSim or any other work that addresses the problem), the experimental results really stand alone and it is hard to draw any conclusion on a more general level beyond the very specific chosen setup and numbers.
> >
> > anyways, as long as the authors include MixSim in the related work of the paper, and also give the mentioned arguments why it is not compared against experimentally, i don't see a fundamental problem anymore. so in this conditional sense, i'm switching to a "yes" re. "claims and evidence".

---

> > > ### Comment · Reviewer_eN7B · 2024-12-19
> > >
> > > addition:
> > >
> > > i briefly checked the updated manuscript. i saw that the authors did include MixSim briefly in the related works, which is good. but i think it should be made more clear what precisely the supposed difference is. i agree that MixSim experiment section is a bit short on some details, but the way i get it is that they also condition on the ground truth path/route, which would be similar to what the submission under discussion does, if i understand correctly. so i'd kindly ask the authors do be more specific in this regard in the manuscript, based on the authors arguments given above.

---

> > > > ### Author Response · Authors · 2024-12-20
> > > > **Re: Official Comment by Reviewer eN7B**
> > > >
> > > > We would like to thank the reviewer for taking the time to review our revised manuscript and for responding to our follow-up. We have updated the related work section to include all the points mentioned in our reasoning above, clarifying how MixSim differs from our work. In general, we believe that our work complements MixSim rather than competes with it. Specifically, our main contribution—the proposed training sampling scheme for conditions—could enhance MixSim's training setup, assuming the authors implemented a similar approach in that regard.

---

### Review · Reviewer_N7Us · 2024-12-05

**Summary Of Contributions:**

This paper extends ITRA by introducing Control-ITRA, a framework designed to train conditioned driving models. Control-ITRA enables precise control of agent behavior through two key mechanisms: specifying waypoints for the agent to follow and setting a target speed for the agent to achieve.

**Audience:**

Yes

**Claims And Evidence:**

Yes

**Requested Changes:**

Please see the weakness part.

**Strengths And Weaknesses:**

### Strengths

- The paper addresses a significant challenge for Level 5 self-driving cars: learning a conditioned multi-agent driving model for controllable simulation design.

- The evaluation is comprehensive, covering multiple benchmarks and metrics.

- The proposed framework is flexible and can be extended to accommodate other conditions and design requirements beyond waypoint-following and aggressiveness.

### Weaknesses

- The clarity of the background on ITRA can be enhanced by providing more detailed explanations of the notations.

- The baselines could be improved by including more recent works, particularly those introduced in the second part of Section 2 that leverage conditional generative models. For instance, in TorchDriveEnv, only basic RL algorithms are considered as baselines, making it difficult to demonstrate the state-of-the-art performance of the proposed method.

- As highlighted in the paper, the strategy for selecting training conditions is critical. Ablation studies comparing alternative strategies for condition selection would strengthen the evaluation.

- Additional visualizations, such as videos, would be valuable for showcasing the simulated driving behaviors.

- Please consider publishing the codebase.

---

> ### Author Response · Authors · 2024-12-18
> **Response to Reviewer N7Us**
>
> We sincerely thank the reviewer for their encouraging and helpful feedback that will help improve our paper.
>
> - **Regarding the comment on "The clarity of the background on ITRA can be enhanced by providing more detailed explanations of the notations":**
>
> We appreciate this feedback and would be grateful if the reviewer could kindly point out specific notations in the ITRA background section that would benefit from additional clarification. This would help us ensure our revisions address all areas of concern effectively.
>
> - **Regarding the suggestion that "The baselines could be improved by including more recent works, particularly those introduced in the second part of Section 2 that leverage conditional generative models":**
>
> Thank you for suggesting the inclusion of more recent works as baselines, particularly those using conditional generative models. We should clarify that our primary contribution focuses on two aspects: (1) an improved sampling scheme for selecting ground truth conditions during training, and (2) a novel approach to incorporating waypoints and target speed conditions into an existing unconditional driving model (ITRA). Our framework enables flexible conditioning, where some agents can be conditioned while others remain unconditional. While we don't claim state-of-the-art performance for the base driving model, we chose ITRA because its per-agent world representation is particularly suitable for applying our conditioning approach. We believe our proposed training goal sampling scheme could benefit any goal-conditional method from the literature that employs similar model training.
>
> - **Regarding the comment that "Ablation studies comparing alternative strategies for condition selection would strengthen the evaluation":**
>
> We appreciate this valuable suggestion. In our current evaluation, we focused on testing what we considered to be the most relevant strategies for training conditional models in this setting. We would welcome specific suggestions for additional strategies that the reviewer believes would provide meaningful comparisons and strengthen our analysis.

---

### Decision · Action_Editor_LxkM · 2025-01-13

**Recommendation:** Reject

**Comment:**

N7Us: "The evaluation results are not sufficient, and the codebase is not published."

eN7B: "overall though, i see the contribution of this paper as rather limited. also i think the paper can/should still improve on how the relation to existing work is discussed, given that established existing work already addresses similar tasks, but negotiating the formulation of individual sentences is beyond my task as reviewer (see also my conversation with the authors). but i don't see a fundamental issue anymore"

bsPZ:  The most positive reviewer, also the most junior.

**Audience:**

While the paper's findings would likely be of interest to the researchers working on machine learning for autonomous driving, robotics, and related areas, the specialized focus and the reviewers' concerns about the evaluations and baselines could limit the appeal to the entire TMLR community.

**Claims And Evidence:**

The paper introduces Control-ITRA, which influences simulated driving agents to use waypoint assignment and target speed modulation. The authors demonstrate that Control-ITRA can control driving behavior using waypoints and target speeds while maintaining realism and achieving good performance both in-domain and out-of-domain.

While the authors provide evidence to support their claims, the reviewers raised valid concerns about the comprehensiveness of the evaluations and the limitations of the baselines. The evidence presented suggests that Control-ITRA can follow waypoints and target speeds, and maintains realistic driving behavior to some extent, but the lack of comparisons to relevant state-of-the-art methods like MixSim and the use of simple RL baselines in some experiments weaken the overall strength of the claims. Although the authors addressed many of these concerns, the initial feedback highlights the need for more robust evaluations and comparisons to establish the true contribution of their method, which was left unaddressed. The claims are therefore not fully supported with entirely sufficient evidence.

I would encourage the authors to majorly revise the paper, and resubmit with the additional baselines and stronger related works section.

**Resubmission Of Major Revision:**

The authors may consider submitting a major revision at a later time.